# Study on Weight Function Distribution of Hybrid Gas-Liquid Two-Phase Flow Electromagnetic Flowmeter

**DOI:** 10.3390/s20051431

**Published:** 2020-03-05

**Authors:** Yulin Jiang

**Affiliations:** College of Information Engineering, Zhongshan Polytechnic College, Zhongshan 528404, China; ghostjiang@sina.com

**Keywords:** gas-liquid two-phase flow, “source” and “field”, Green’s function, superposition property, weight function

## Abstract

The electromagnetic flowmeter is usually used for single-phase fluid parameter measurement. When the measured fluid is gas-liquid two-phase flow, the geometry of the sensor measurement space will change with the movement of the gas, which will cause measurement errors. The weight function distribution is an important parameter to analyze such measurement errors. The traditional method for calculating the weight function of gas-liquid two-phase flow involves complex dimensional space transformation, which is difficult to understand and apply. This paper presents a new method for calculating the weight function of the gas-liquid two-phase flow electromagnetic flowmeter. Firstly, based on the measurement principle of the electromagnetic flowmeter, a general model of weight function of the gas-liquid two-phase flow electromagnetic flowmeter is built. Secondly, the bubbles in the fluid are regarded as the “isolated” points in the flow field. According to the physical connection between the “field” of the measured fluid and the “source” of the sensor electrode, the Green’s function expression based on gas-liquid two-phase flow is established. Then, combined with the boundary conditions of the measurement space of the electromagnetic flowmeter, the Green’s function is analyzed. Finally, the general model of weight function is solved by using the expression of Green’s function, then the expression of the weight function of the electromagnetic flowmeter is obtained when the measured fluid is hybrid gas-liquid two-phase flow. The simulation results show that the proposed method can reasonably describe the influence of the gas in the measured fluid on the output signal of the sensor, and the experimental results also indirectly prove the rationality of this method.

## 1. Introduction

The electromagnetic flowmeter is a meter based on Faraday’s law of electromagnetic induction for measuring the average flow velocity of conductive fluids. Because it has the advantages of high measurement accuracy, wide measuring range, low power consumption, and the measurement results are independent of the viscosity and density, it is widely used in single-phase flow parameter measurement [1]. However, in chemical, food, and nuclear reactor engineering, fluids are inevitably doped with gas, which makes the geometry of the measurement space of the electromagnetic flowmeter change dynamically. According to Faraday’s law of electromagnetic induction, the output signals of electromagnetic flowmeter are mainly related to the flow field distribution, magnetic field distribution, and weight function [2,3,4]. When the fluid in the measuring pipeline has fully developed, the flow field distribution and magnetic field distribution no longer change, then the weight function becomes the key factor affecting the output signal of the flowmeter. In the field of electromagnetic flowmeter measurement, the weight function is used to describe the contribution of fluid microelements in the pipeline to the output signal. The weight function is only related to the geometry of the measurement space of the electromagnetic flowmeter [5,6]. It reflects the degree of influence of the sensor electrode installation position, electrode shape, flow field distribution, etc., on the induced potential. In the hybrid gas-liquid two-phase flow, since the bubbles in the fluid change the geometry of the measurement space, the distribution of the weight function also changes. The sensor structure with good response characteristics has a uniform weight function distribution and is less affected by bubbles. Therefore, the distribution of the weight function is also an important factor reflecting the rationality of the sensor structure. Especially in the “dry” calibration method of electromagnetic flowmeter [6,7], the distribution of the weight function directly determines the rationality of the sensor structure and the feasibility of the “dry” calibration method. Therefore, the weight function distribution of the hybrid gas-liquid two-phase flow electromagnetic flowmeter has been a research focus in the measurement field.

Since the weight function is an important parameter in the research field of electromagnetic flowmeters, a series of studies have been performed on the weight function in the past few decades. Shercliff J. A. [2] first proposed the concept of weight function of the electromagnetic flowmeter in 1962. Bavir et al. [3] used the virtual current method to solve the weight function of the single-phase flow electromagnetic flowmeter. Wei Kaixia et al. [8] used an ansys finite element simulation method to study the weight function distribution of partially-filled electromagnetic flowmeters in the case of gas-liquid separation and gave the simulation results. Yin Shiyi et al. [9] used a resistance network method to theoretically analyze the weight function distribution of partially-filled pipe electromagnetic flowmeter under gas-liquid separation, and gave the weight function distribution formula for a multi-electrode sensor. Li Xuejing et al. [10,11] studied the weight function of the non-insulated and partially-filled pipe electromagnetic flowmeter by using Comsol finite element simulation method and discussed the influence of the non-insulated pipe wall on the sensor output signal. All these studies assume that the gas and liquid in the measured fluid are separated, and the gas does not affect the output signal of the electrode sensor. However, in the transmission of pulp and food sauce, the gas is doped in the fluid, and the position of the gas also changes with the flow of the fluid, and the output of the sensor changes significantly. For such fluid, Shi Yanyan et al. [12] used a conformal transformation method to analyze the weight function of electromagnetic flowmeters using arc electrodes. However, the arc-shaped electrode is easily contaminated by the measured fluid, thereby reducing the measurement accuracy. In addition, this method involves complex multi-dimensional transformations and inverse transformations, which put forward higher requirements for engineering equipment and engineering personnel, and is not conducive to the application of engineering practice. For the weight function of the electromagnetic flowmeter with point electrode and hybrid gas-liquid two-phase flow, the research results are much less. 

This paper presents a new method for solving the weight function of the gas-liquid two-phase flow. Firstly, the basic measurement equation of the gas-liquid two-phase flow of the electromagnetic flowmeter was derived by using the Maxwell electromagnetic equation. Secondly, the Green’s function [13,14,15] was used to construct the weight function model of the gas-liquid two-phase flow. Finally, using the superposition property of Green’s function and the boundary conditions of the sensor’s measurement space, the weight function model is solved, and the expression of the weight function of the electromagnetic flowmeter is obtained when the measurement fluid is hybrid gas-liquid two-phase flow. The simulation results show that the weight function method proposed in this paper is simple and easy to understand, and can well describe the degree of influence of the gas in the gas-liquid two-phase flow on the measurement results, and has a strong reference value for the design and installation of the sensor structure.

## 2. Measurement Principle of Gas-Liquid Two-Phase Flow Electromagnetic Flowmeter

In this part, the measurement principle of gas-liquid two-phase flow electromagnetic flowmeter is introduced, and the general model of weight function of gas-liquid two-phase flow electromagnetic flowmeter is established.

### 2.1. Basic Measurement Equation

When the fluid of the electromagnetic flowmeter is a gas-liquid two-phase flow, the structure diagram of the electromagnetic flowmeter is shown in Figure 1.

Where R is the radius of the measured flowmeter pipe, $R_{1}$ is the radius of bubble, the shape of the exciting coil is saddle-shaped, and it is close to the outside of the tube wall. When the current flows through the excitation coil, a certain intensity of electromagnetic field will be generated in the measurement space of the electromagnetic flowmeter, and the direction of the magnetic field intensity B is perpendicular to the plane formed by the electrode and the axis. According to Faraday’s law of electromagnetic induction, when fluid flows through the measured flowmeter pipe, corresponding induced electromotive force is generated on the two electrodes of the flow meter [16,17,18]. According to the continuous equation of charge conservation, at any point of the two electrodes connection, the relationship between the current density vector and the charge density is as follows:(1)$$\nabla \cdot \bar{j}+\frac{\partial \rho }{\partial t}=0$$
where $\bar{j}$ is the imaginary current density vector, ρ is the bulk charge density, t is instantaneous time unit, and $\frac{\partial \rho }{\partial t}$ is the instantaneous time change rate of the bulk charge density. According to Maxwell’s Equation, $div\bar{D}=\rho$, and $\bar{D}$ is the electric displacement vector, then Equation (1) can be expressed as:(2)$$\nabla \cdot \bar{j}+\nabla \cdot \frac{\partial \bar{D}}{\partial t}=0$$

When the fluid in the pipeline flows, the magnitude of the virtual current between the two electrodes can be expressed as:(3)$$\bar{j}=\sigma (\bar{E}+\bar{v}\times \bar{B})$$
where $\bar{E}$ is the electric field intensity generated by the electrode and applied to the charge, σ is the conductivity of the fluid in the pipeline, $\bar{v}$ is the flow velocity of the fluid, and $\bar{B}$ is the magnetic field strength generated by the excitation coil. When the displacement current can be ignored, the divergence of the virtual current is zero; that is: (4)$$\nabla \cdot \bar{j}=\nabla \cdot \sigma (\bar{E}+\bar{v}\times \bar{B})=0$$

Assume the magnitude of fluid conductivity is uniform. Let U be the potential difference between the two electrodes; that is $\bar{E}=-\nabla U$. then Equation (4) can be simplified as
(5)$$\nabla^{2}U=\nabla \cdot (\bar{v}\times \bar{B})$$

Equation (5) is the basic measurement equation of the electromagnetic flowmeter.

### 2.2. General Model for Weight Function of Gas-Liquid Two-Phase Flow

Assuming that the tube wall and the bubble in the tube are insulators, this means there is no normal virtual current on the boundary; that is ${\bar{j}}_{n}=0$. Because the fluid velocity is zero at the tube wall boundary, then the normal derivative of the potential difference can be derived from $\bar{E}=-\nabla U$ and $\bar{j}=\sigma \bar{E}$, which has the following relations:(6)$$\frac{\partial U}{\partial n}=0$$

With the help of Green’s function G and corresponding boundary conditions, the induced electromotive force represented by Equation (5) can be solved. For the inner tube wall and bubble surface with surface insulation, the partial derivative of the Green’s function is ∂G/∂n=0. For the two electrodes on the tube wall, it can be regarded as the impulse function caused by the unit virtual current flowing through the two electrodes. Then, the partial derivative of Green’s function on two electrodes can be expressed as:(7)$$\frac{\partial G}{\partial n}=\left\{ \begin{array}{cc} +\frac{2(R+R_{1})L\delta (\phi )}{S_{0}} & \;electrode1\\ -\frac{2(R+R_{1})L\delta (\phi )}{S_{0}} & \;electrode2\\ 0 & the\;reset \end{array}\right.$$
where $S_{0}$ is the surface area of the electrode exposed in the fluid, L is the half-length of the effective pipeline length of the flow meter, and δ(ϕ) is the impulse function generated when the virtual current in the pipe flows through the electrode sensor. Because the Green’s function satisfies the Laplace equation, then:(8)$$\nabla^{2}G=0$$

Assume that the induced potential generated by electrode 1 is $U_{1}$, the induced potential generated by electrode 2 is $U_{2}$, and the induced potential difference between the two electrodes is $U_{12}$; then, according to the second formula of the Green’s function [19,20], we can get:(9)$$\begin{array}{l} \int (U\nabla^{2}G-G\nabla^{2}U)dV\\ =\oint (U\frac{\partial G}{\partial n}-G\frac{\partial U}{\partial n})d\sum_{S}\\ =2(R+R_{1})L(U_{1}-U_{2})\\ =2(R+R_{1})LU \end{array}$$

Then, the induced electromotive force difference between the two sensor electrodes can be expressed as:(10)$$\begin{array}{l} U_{12}=\frac{1}{2(R+R_{1})L}\int (U\nabla^{2}G-G\nabla^{2}U)dV\\ =-\frac{1}{2(R+R_{1})L}\int \nabla \cdot G(\bar{v}\times \bar{B})-\nabla G\cdot (\bar{v}\times \bar{B})dV\\ =\frac{1}{2(R+R_{1})L}\int \bar{v}\cdot (\bar{B}\times \nabla G)dV \end{array}$$

Assuming that the fluid in the pipeline is a stable axisymmetric flow, and $v_{x}=v_{y}=0,v_{z}=v$, then:(11)$$\begin{array}{l} \bar{v}\cdot (\bar{B}\times \nabla G)=v_{x}\cdot (\bar{B}\times \nabla G{)}_{x}+v_{y}\cdot (\bar{B}\times \nabla G{)}_{y}+v_{z}\cdot (\bar{B}\times \nabla G{)}_{z}\\ =v_{z}(B_{x}\frac{\partial G}{\partial y}-B_{y}\frac{\partial G}{\partial x})e_{z} \end{array}$$

Suppose x=rcosϕ,y=rsinϕ(r∈[0,R]), and replace the relevant part in Equation (10) with Equation (11), then Equation (10) can be further expressed as:(12)$$\begin{array}{l} U_{12}=\frac{1}{2(R+R_{1})L}\int \bar{v}\cdot (\bar{B}\times \nabla G)dV\\ =-\frac{1}{\left(R+R_{1}\right)}\int_{R_{1}}^{R}\int_{0}^{2\pi }v_{z}(r,\phi )B_{y}[\cos\phi \frac{\partial G(r,\phi )}{\partial r}-\frac{\sin\phi }{r}\frac{\partial G(r,\phi )}{\partial \phi }]rdrd\phi \end{array}$$

The Equation (12) is the expression of the induced electromotive force of the gas-liquid two-phase flow electromagnetic flowmeter.

If the measured fluid is stable axisymmetric flow, it is assumed that the magnetic field is a uniform magnetic field and perpendicular to the plane direction formed by the electrode line and the z-axis, that is $B=B_{x},B_{y}=0$, then Equation (12) can be further simplified as:(13)$$U_{12}=\frac{1}{\left(R+R_{1}\right)}\int_{R_{1}}^{R}v_{z}(r)B_{x}\int_{0}^{2\pi }\sin\phi \frac{\partial G(r,\phi )}{\partial r}+\frac{\cos\phi }{r}\frac{\partial G(r,\phi )}{\partial \phi }d\phi rdr$$
where
(14)$$W=\sin\phi \frac{\partial G(r,\phi )}{\partial r}+\frac{\cos\phi }{r}\frac{\partial G(r,\phi )}{\partial \phi }$$

Equation (14) is the general model of the weight function of the gas-liquid two-phase flow electromagnetic flowmeter. Then, the induced potential difference between the two electrodes of the sensor can be expressed as:(15)$$U_{12}=\frac{1}{\left(R+R_{1}\right)}\int_{R_{1}}^{R}v_{z}(r)B_{x}\int_{0}^{2\pi }W(r,\phi )d\phi rdr$$

According to Equation (15), the sensor’s induced electromotive force is related to the velocity v of the fluid flowing through the pipe, the distribution of the magnetic field intensity $B_{x}$, and the distribution of the weight function W. The weight function is used to describe the contribution of the fluid microelements in the pipeline to the induced potential of the electrode, and its value is only related to the structure of the sensor. Obviously, in the case where the flow field is fully developed and the magnetic field is a uniform magnetic field, the distribution of the weight function directly affects the output of the sensor’s induced potential. 

## 3. Solution of Weight Function Model

In this part, the relationship between bubbles and flow fluid will be further described from the perspective of “Source” and “Field”, and then the weight function model is solved by using the properties of Green’s function and the boundary conditions of sensor measurement space.

### 3.1. Description of “Source” and “Field”

According to the previous analysis, the weight function is a direct factor affecting the induced potential of the gas-liquid two-phase flow electromagnetic flowmeter, and the weight function is determined by the Green’s function in the calculation process. Therefore, the weight function model of gas-liquid two-phase flow is studied from the physical meaning of Green’s function in this paper. From a physical point of view, the electrode-induced electromotive force can be regarded as the “source” of the induced electromotive force of each fluid microelement in the effective field of the electromagnetic flowmeter, and the degree of influence of the induced potential of each fluid microelement on the “source” can be regarded as a "field”. When the electrode induced electromotive force is decomposed into the superposition of several “point sources”, if the “field” generated by the “point source” can be known, using the superposition principle, the “field” of any “source” under the same boundary conditions can be obtained, the “field” generated by the “point source” is called the Green’s function.

The weight function is used to describe the degree of contribution of the fluid in the pipeline to the electrode induced electromotive force. The electrode induced electromotive force can be regarded as the total contribution of the partial contribution of the fluid in the pipeline. In the gas-liquid two-phase flow electromagnetic flowmeter, the induced electromotive force of the electrodes can be viewed as the superposition of “sources” generated by two different “fields” under the same boundary conditions, that is the “sources” generated by the measured fluid overlay with the “source” created by the bubbles. Based on this idea, the model of weight function can be further solved.

### 3.2. Solution of the Model

According to the superposition property of the Green’s function, the Green’s function of the gas-liquid two-phase flow can be expressed as:(16)$$G((z-z_{0})|z^{\prime }))=G(z_{c}|z^{\prime })$$
where $G(z_{0})$ is the reverse contribution of the bubble part, z represents the point in the measurement pipe, $z_{0}$ represents the point in the bubble, $z^{\prime }$ represents the sensor electrode, and $z_{c}$ represents the point in the measuring pipe except the bubble. Assuming that the offset of the center of the bubble relative to the center of the pipe is (a,b), then the rectangular coordinate expression of the bubble can be expressed as:(17)$${\left(x-a\right)}^{2}+{\left(y-b\right)}^{2}=R_{1}^{2}$$
and the polar coordinate expression for any point inside the bubble can be expressed as:(18)$$z_{0}=r_{0}e^{i\phi }+ke^{i\theta }\;\left(0\leq r_{0}\leq R_{1}\right)$$
where $k=\sqrt{a^{2}+b^{2}}$, θ=arctan(b/a), the angle of the two electrodes relative to the center of the bubble can be expressed as:(19)$$\left\{ \begin{array}{l} \phi^{\prime }=\frac{\pi }{2}+\alpha \;electrode1\\ \phi^{\prime }=\frac{3\pi }{2}-\beta \;electrode2 \end{array}\right.$$
where $\alpha =arctg(\frac{a}{R-b})$, $\beta =arctg(\frac{a}{R+b})$. Then, the normalized expression of the Green’s function can be expressed as:(20)G(zc|z′)=Re[−ln(|zc−z′|2)]=−ln(|rceiϕ−keiθ−eiϕ′|2)=−12πln[rc2−2rccos(ϕ−ϕ′)+1−2keiθ(rccosϕ−cosϕ′−keiθ)]

When the bubble in the pipe is at the eccentric position, the model of the weight function can be expressed as:(21)$$W=\sin\phi \frac{\partial G(r_{c},\phi )}{\partial r_{c}}+\frac{\cos\phi }{r_{c}}\frac{\partial G(r_{c},\phi )}{\partial \phi }$$
where:(22)$$\frac{\partial G(r_{c},\phi )}{\partial r}=-\frac{1}{\pi }\frac{r_{c}-\cos(\phi -\phi^{\prime })-ke^{i\theta }\cos\phi }{r_{c}^{2}-2r_{c}\cos(\phi -\phi^{\prime })+1-2ke^{i\theta }(r_{c}\cos\phi -\cos\phi^{\prime }-ke^{i\theta })}$$
and:(23)$$\frac{\partial G(r_{c},\phi )}{\partial \phi }=-\frac{1}{\pi }\frac{r_{c}\sin(\phi -\phi^{\prime })+ke^{i\theta }\sin\phi }{r_{c}^{2}-2r_{c}\cos(\phi -\phi^{\prime })+1-2ke^{i\theta }(r_{c}\cos\phi -\cos\phi^{\prime }-ke^{i\theta })}$$

Then, the weight function can be further simplified as:(24)$$W=-\frac{1}{\pi }\frac{r_{c}\sin\phi -\sin\phi^{\prime }}{r_{c}^{2}-2r_{c}\cos(\phi -\phi^{\prime })+1-2ke^{i\theta }(r_{c}\cos\phi -\cos\phi^{\prime }-ke^{i\theta })}$$

Substituting Equation (24) into Equation (15), the expression of the electrode-induced electromotive force was obtained when the bubble is eccentric:(25)$$\begin{array}{l} U_{12}=\frac{1}{\pi (R+R_{1})}\\ \int_{R_{1}}^{R}v_{z}(r)B_{x}\int_{0}^{2\pi }\frac{r_{c}\sin\phi -\sin\phi^{\prime }}{r_{c}^{2}-2r_{c}\cos(\phi -\phi^{\prime })+1-2ke^{i\theta }(r_{c}\cos\phi -\cos\phi^{\prime }-ke^{i\theta })}d\phi rdr \end{array}$$
where $v_{z}$ is the fluid velocity distribution in the pipeline. When the bubble size in the gas-liquid two-phase flow is zero, Equations (24) and (25) can be simplified as:(26)$$W=\frac{\sin\phi^{\prime }-r\sin\phi }{1-2r\cos(\phi -\phi^{\prime })+r^{2}}$$
(27)$$U_{12}=\frac{1}{\pi R}\int_{0}^{R}v_{z}(r)B_{x}\int_{0}^{2\pi }\frac{\sin\phi^{\prime }-r\sin\phi }{1-2r\cos(\phi -\phi^{\prime })+r^{2}}d\phi rdr$$

Equations (26) and (27) are weight function expressions and induced potential expressions of full-tube single-phase flow electromagnetic flowmeters.

## 4. Simulation and Analysis

In this part, the uniformity of weight function is defined, and the comsol multiphysics simulation software was used to simulate the weight function Expression (24) to verify the feasibility of the proposed method. 

### 4.1. Definition of Weight Function Uniformity

According to the weight function theory, the more uniform the weight function distribution, the less the sensor’s induced potential is affected by the velocity distribution. The expression of the uniformity for the discretization of the weight function is defined as follows:(28)$$\eta =\frac{1}{2\pi R^{2}N}\sum_{i=1}^{N}\left|\frac{W_{i}-\bar{W}}{\bar{W}}\right|$$
where N is the number of discrete points divided by the weight function, $W_{i}$ is the weight function value of a single discrete point, and $\bar{W}$ is the uniform value of the weight function:(29)$$\bar{W}=\frac{1}{\pi R^{2}N}\sum_{i=1}^{N}W_{i}$$

In Formula (28), the parameter η reflects the overall uniformity of the weight function distribution in the pipeline. The smaller the value of η is, the better the uniformity of the weight function distribution is, and the less the induced potential of the sensor is affected by the velocity distribution. At the same time, in the case of gas-liquid two-phase flow, the uniformity of the weight function can also be used to measure the degree of influence of bubbles on the measurement results.

### 4.2. The “source” and “field” of the Weight Function

It can be known from the foregoing description that the total amount of induced potential of the sensor electrode is “source”, and the induced potential generated by the fluid cutting magnetic field of each part in the pipeline is “field”. The relationship between these two parameters can be described by the virtual field shown in Figure 2. The conversion relationship between the two parameters is the weight function distribution. The comsol multiphysics software was used to solve Equation (26) of the weight function, and the distribution of the weight function was obtained. Then, the weight function value $W_{center}$ of the pipe center is taken as the normalized base, and normalization processing is performed to obtain the normalized weight function distribution, as shown in Figure 3. The following gas-liquid two-phase flow weight function distributions are normalized based on $W_{center}$.

The upper and lower ends of Figure 2 and Figure 3 are the two sensor electrodes of the electromagnetic flowmeter. Figure 2 illustrates that each part of the fluid in the pipeline has a certain effect on the sensor’s induced potential, and the weight function distribution shown in Figure 3 indicates the degree of influence of the fluid “field” on the “source” of the sensor’s induced potential.

### 4.3. Comparison of Weight Function Distribution in Concentric

When the bubble and the measurement pipe are just concentric, that is a=b=θ=0, compare the weight function when the bubble size is different. When q = 0.05, 0.1, 0.15, 0.2, the normalized weight function distribution is shown in Figure 4a–d, where q = R_1_/R represents the relative size of the bubble and the diameter of the pipe.

It can be seen from Figure 4 that compared to the distribution of the weight function in the single-phase flow, the weight functions at the upper and lower ends of the bubble are significantly smaller, and the weight functions at the left and right sides are larger. That is, the contribution of the “field” on the left and right sides of the bubble to the “source” of the electrode-induced potential is significantly increased, while the contribution of the “field” above and below the bubble to the “source” of the electrode induced potential is significantly reduced. When the bubble is small, the distribution of the weight function is similar to that of single-phase flow, which indicates that when the bubble is small, the bubble has little effect on the measurement result of the electromagnetic flowmeter. However, when the bubble gradually increases, the local uniformity of the weight function changes greatly. The fluid weight in the vertical direction near the bubble becomes smaller, while the weight in the left and right direction becomes larger, but the weight is still much smaller than that near the electrode. This shows that the magnitude of the induced electromotive force of the electrode is increasingly dependent on the flow field around the electrode.

According to Equation (28), the uniformity of the weight function and the relative variation of the uniformity can be obtained, as shown in Table 1.

Obviously, when the center of the bubble coincides with the center of the pipe, the uniformity of the weight function decreases with the increase of the bubble, indicating that the output signal of the sensor electrode is less affected by the flow field distribution. However, with the increase of the bubble, the relative change rate of the weight function is larger and larger compared with the single-phase flow, which indicates that the presence of the bubble has more and more influence on the output signal of the sensor.

### 4.4. Comparison of Weight Function Distributions at Eccentricity

When the bubble and the measurement pipe are in an eccentric state, suppose that the center of bubble is (a,b), and $e=\sqrt{a^{2}+b^{2}}/R$ presents the degree of eccentricity. When the bubble size q = 0.2, the weight function of the bubble at different positions on the horizontal axis of the pipeline (i.e., b = 0) is compared, and the distribution of the weight function is shown in Figure 5a–d. Because the sensor electrodes are installed on both sides of the pipe, when the bubble is on the horizontal axis, it is exactly on the same line with the electrode.

As shown in Figure 5, when the bubble and the two electrodes are located on the same line, the weight function near the upper and lower electrodes all changes. The closer the bubble is to the electrode, the more the weight function near the electrode changes, which indicates that the fluid near the electrode has more and more influence on the output signals of the sensor. In contrast, the weight function near the other electrode changes more and more gently, indicating that the presence of bubbles significantly changes the distribution of the influence of the fluid on the sensor output signal.

Similarly when the bubble size q = 0.2, the weight function of the bubble at different positions on the vertical axis of the pipeline (i.e., a = 0) is compared, and the distribution of the weight function is shown in Figure 6a–d. 

It can be seen from Figure 6 that when the bubble is located on the vertical axis of the pipe, the closer it is to the pipe wall, the more the local uniformity of the weight function changes near the bubble, but the weight function on the other side approaches the weight function distribution of the single-phase flow, which shows that the effect of bubbles on the output signal of the sensor electrode is gradually decreasing.

According to Equation (28), the uniformity of the weight function can be obtained when the bubble size is q = 0.2, and the bubbles are located in the vertical axis and the horizontal axis of the pipeline with different eccentricities, respectively. Compared with the uniformity of weight function of the single-phase flow, the relative change rate is shown in Figure 7.

Obviously, when bubbles are doped in the fluid of the electromagnetic flowmeter, the uniformity of the weight function changes significantly. When the bubble is located on the vertical axis of the pipe and is perpendicular to the electrode connection direction, the relative change rate of the uniformity of the weight function decreases as the eccentricity increases. The closer to the pipe wall, the smaller the uniformity, which indicates that the influence of bubbles on the output signal of the sensor is smaller. 

When the bubble is located on the horizontal axis of the pipe, the relative change rate of uniformity presents non-monotonic change. It can be seen from Figure 5 that, compared with the distribution of the weight function in the single-phase flow, the weight functions at the upper and lower ends of the bubble are significantly smaller, and the weight functions at the left and right are larger. When the bubble approaches the electrode from the center of the pipe, the weight function of the lower side gradually increases, while the weight function of the upper side decreases relatively, and the uniformity of the whole weight function decreases. As the bubble approaches the electrode, the weight function of the lower part of the pipe gradually tends to the weight function in the single-phase flow, and near the upper electrode, the weight function changes more and more drastically due to the existence of the bubble, and the uniformity of overall weight function also gradually increased. So when the bubble approaches the electrode, the relative change rate of uniformity decreases first and then increases, indicating that the effect of the bubble on the sensor output signal decreases first and then increases.

## 5. Experiment and Analysis

Since the weight function is a parameter that cannot be measured directly, it can be known from Equation (15) that when the magnetic field distribution and velocity distribution are constant, the induced potential difference of sensor is only related to the weight function distribution. In this section, we obtain the weight function distribution indirectly by measuring the induced potential output from the sensor of the electromagnetic flowmeter, thereby verifying the rationality of the weight function calculation method proposed in this paper.

### 5.1. Experimental Equipment

The experimental equipment is shown in Figure 8. The fluid flows from the upper water tank, and the fluid flow rate is controlled by the flow control valve. The instrument accuracy of the standard flowmeter is 0.5 level, and the error of the indicated value is 0.152%. The vertical part of the pipe is a self-made measuring pipe. The inner diameter of the pipe is 30 mm, the wall thickness is 3 mm, and the material of the measuring pipe is transparent acrylic pipe. The self-made electromagnetic flowmeter is installed vertically on the water pipe, and the output signal of the sensor electrode can be displayed on the oscilloscope in real-time after being processed. The bubbles in the vertical pipeline are generated by the gas storage tank. The release valve is used to control the strength of the gas flow, thereby generating two-phase flows with different gas holdups. The retractable pipe at the front is controlled to adjust the position of the bubble output, and the distance from the bubble output port to the electromagnetic flowmeter is 700 mm. The pressure sensor A is installed at the position of the bubble output port, the pressure sensor B is located above the self-made flowmeter, and the distance between A and B is 1116 mm. Three high-speed cameras are installed at the lower end of the self-made flowmeter to take real-time photos of the flow field from three different angles, and the photo frequency is 100 frames/s. By analyzing the camera picture, the size and position of the bubble within the measured interval can be determined.

Because the size of the bubbles generated by the gas tank device is not consistent in different time periods, in the actual calculation, the average value of the sampled gas bubbles per unit time was used to instead of the bubble size. Then the size of the bubble can be expressed by the following Equation:(30)$$V_{av}=\sum_{i=1}^{k}V_{i}$$
where $V_{av}$ is the average value of bubble volume per unit time, k is the average number of bubbles sampled per unit time, and $V_{i}$ is the volume of sampled bubbles. When the fluid in the pipe is turbulent, the flow of the fluid is in a disordered state, and the position of the bubble in the section of the pipe is not fixed, as shown in Figure 9:

Therefore, during the experiment, the measurement was repeated many times, and the average value of the bubble position in the unit time was used to replace the bubble positions. Then, the expression of the bubble position is:(31)$$P_{av}(x,y)=\frac{1}{k}\sum_{j=1}^{k}\left(\frac{1}{3}\sum_{i=1}^{3}P(x_{i,j},y_{i,j})\right)$$
where $P(x_{i,j},y_{i,j})$ is the position of the j-th bubble on the i-th camera. Similarly, the average gas holdup in the measurement interval can be expressed as:(32)$$\varepsilon =\frac{p_{B}-p_{A}}{\rho g\Delta H}$$
where ΔH is the distance between the pressure sensors A-B, $p_{A}$ and $p_{B}$ are the average output of the pressure sensor in a unit time, g is the acceleration of gravity, and the sampling frequency of the pressure sensor is 100 Hz.

### 5.2. Experimental Results

During the initial measurement, the gas release valve is closed, and the fluid in the measurement interval is single-phase flow. When the fluid in the horizontal pipeline reaches stable and full development, the flow velocity of the fluid can be obtained from the standard flow meter is 0.56 m/s, and the Reynolds number of the fluid in the measured interval is calculated to be 16,800, then the fluid field is turbulent, and the diagram of velocity distribution is shown in Figure 10. At the same time, it can be observed from the oscilloscope that the potential difference output from the self-made electromagnetic flowmeter sensor is 79.31 mv after signal conditioning, as shown in Figure 11.

Next, open the release valve of the gas storage tank to generate gas bubbles of different sizes and change the gas holdup of the fluid. When the bubble is located around the center of the pipe, the real-time value of the induced potential difference of the sensor can be observed from the oscilloscope, and the average value is taken for multiple measurements. The results are shown in Table 2.

Obviously, when the bubble is small, the effect of the bubble on the flowmeter output signal is very small, but as the bubble increases, the induced potential difference of the sensor becomes significantly smaller, indicating that the contribution of the fluid to the sensor’s induced potential becomes smaller; that is, the uniformity becomes smaller, which is consistent with the variation trend of the uniformity of the weight function shown in Table 1.

Next, keep the gas holdup in the fluid constant, the gas bubbles are located on the vertical line of the plane formed by the electrode and the pipeline axis (y-axis) through a retractable tube connected to the airflow output pipe. Observe the real-time value of the potential difference of the sensor from the oscilloscope, and take the average value after multiple measurements. When the average size of the bubble q = 0.23, the observation results are shown in Table 3.

In the vertical direction of the plane formed by the electrode and the pipeline axis, the closer the bubble is to the pipe wall, the larger the sensor’s induced potential difference is, indicating that the influence of the bubble on the measurement result of the electromagnetic flowmeter is smaller. In other words, the contribution of the fluid at the bubble position to the sensor’s induced potential becomes smaller and smaller, and this trend is consistent with the distribution of the weight function.

Similarly, keep the gas holdup in the fluid and the size of the bubble constant. When the bubble and the two electrodes are located on the same line (x-axis), observe the real-time value of the potential difference of the sensor from the oscilloscope, and take the average value after multiple measurements. The observation results are shown in Table 4.

Between the two electrodes, the gas bubble approaches the electrode sensor from the center of the pipe, and the induced potential difference between the two electrode sensors increases first and then decreases. From Equation (15), it can be known that when the magnetic field distribution is constant, the induced potential difference of sensor is related to the weight function distribution and velocity distribution. From Figure 10, it can be seen that the flow velocity distribution is monotonically decreasing from the center of the pipe to the pipe wall, which also indirectly verifies that the uniformity of the weight function is first reduced and then increased, which is consistent with the trend of the weight function shown in Figure 7.

## 6. Conclusions

In this paper, the weight function distribution of the hybrid gas-liquid two-phase flow electromagnetic flowmeter is studied, and a new calculation method is proposed. Based on the measurement principle of the electromagnetic flowmeter, a general model of the weight function of gas-liquid two-phase flow is constructed. The model is solved by using the superposition property of Green’s function and the boundary conditions of the electromagnetic flowmeter pipeline, and the hybrid gas-liquid two-phase flow weight function theoretical expression is obtained. Finally, the proposed method is verified by simulation and experiments. The main conclusions are as follows:(1)In order to solve the weight function of the gas-liquid two-phase flow electromagnetic flowmeter, the superposition property of the Green function is used as the main solution method, which greatly reduces the difficulty of solution.(2)When the bubble is located in the center of the measurement area, the local uniformity around the bubble becomes larger, but the overall uniformity becomes smaller. The larger the bubble, the greater the impact on the sensor output signal.(3)When the bubble size is fixed, the farther the bubble is from the electrode, the smaller the relative change rate of the uniformity, the smaller the influence of the bubble on the sensor output signal, and vice versa.(4)The method proposed in this paper can better explain the effect of the gas in the hybrid gas-liquid two-phase flow on the output signal of the sensor and provides a theoretical basis for the error analysis of the parameter measurement of the electromagnetic liquid flowmeter. At the same time, it also provides a theoretical basis for the installation position of the sensor and the rationality of the sensor structure.

## Figures and Tables

**Figure 1 sensors-20-01431-f001:**
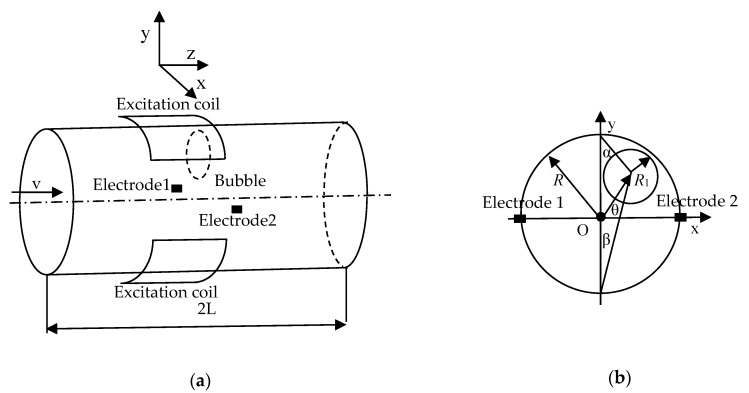
The structure of the electromagnetic flowmeter. (**a**) Three-dimension Diagram; (**b**) Section Diagram.

**Figure 2 sensors-20-01431-f002:**
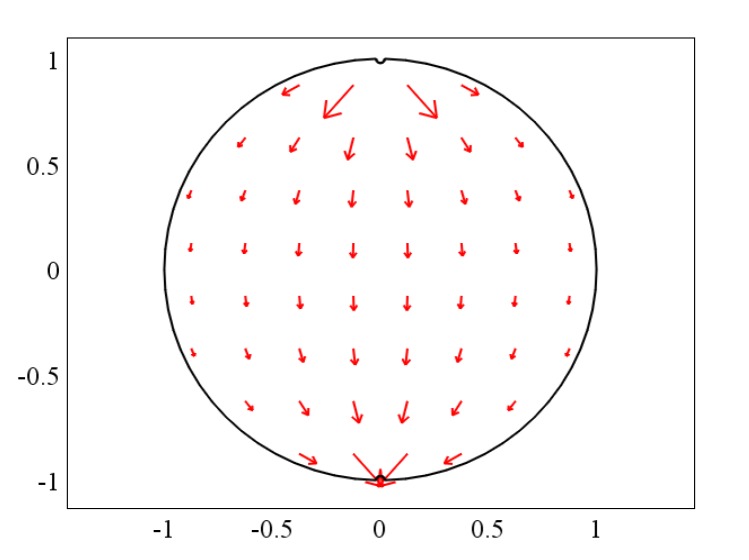
“Source” and “Field” diagrams.

**Figure 3 sensors-20-01431-f003:**
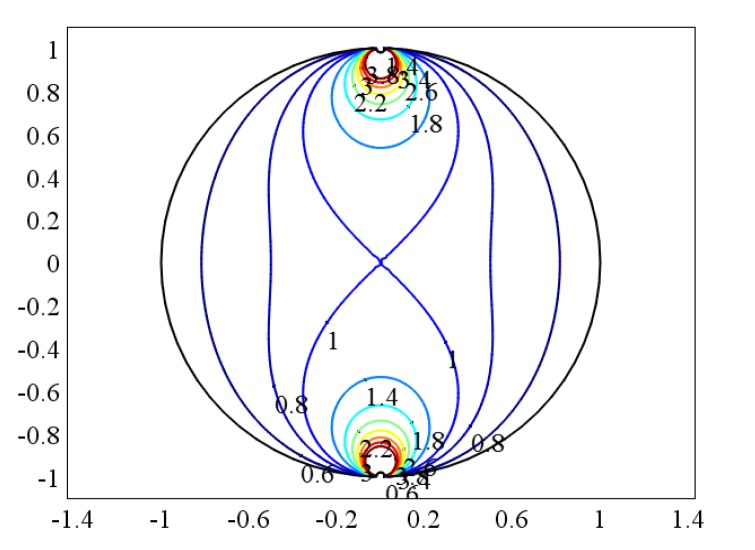
The normalized weight function of the single-phase flow.

**Figure 4 sensors-20-01431-f004:**
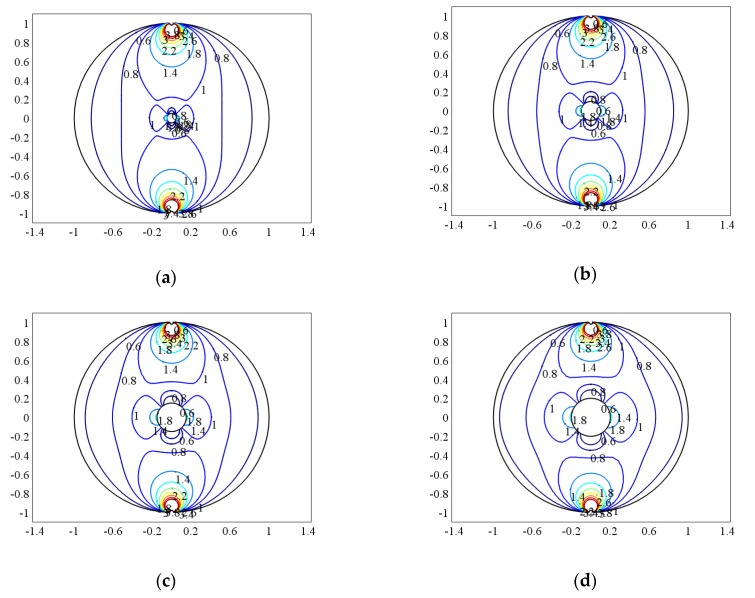
The weight function distribution when the bubble is concentric with the center of the pipe. (**a**) q = 0.05, (**b**) q = 0.1, (**c**) q = 0.15, (**d**) q = 0.2.

**Figure 5 sensors-20-01431-f005:**
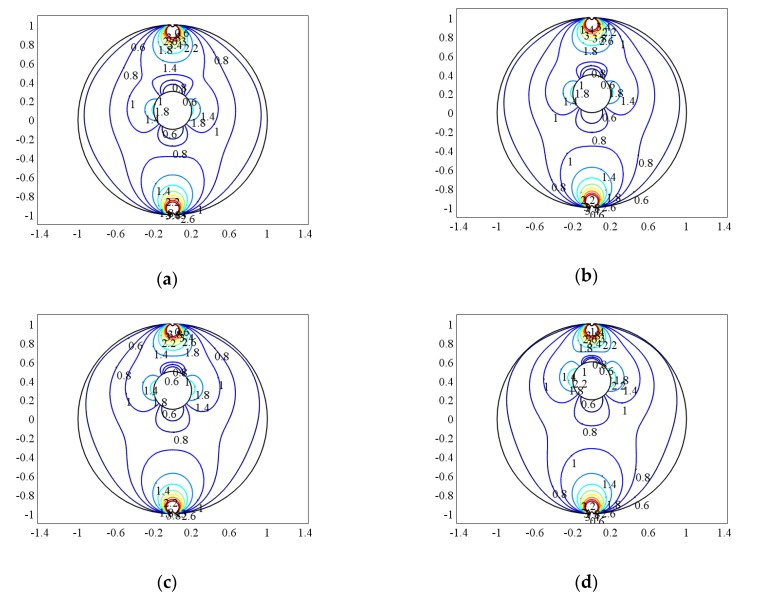
The weight function distribution at b = 0. (**a**) e = 0.1, (**b**) e = 0.2, (**c**) e = 0.3, (**d**) e = 0.4.

**Figure 6 sensors-20-01431-f006:**
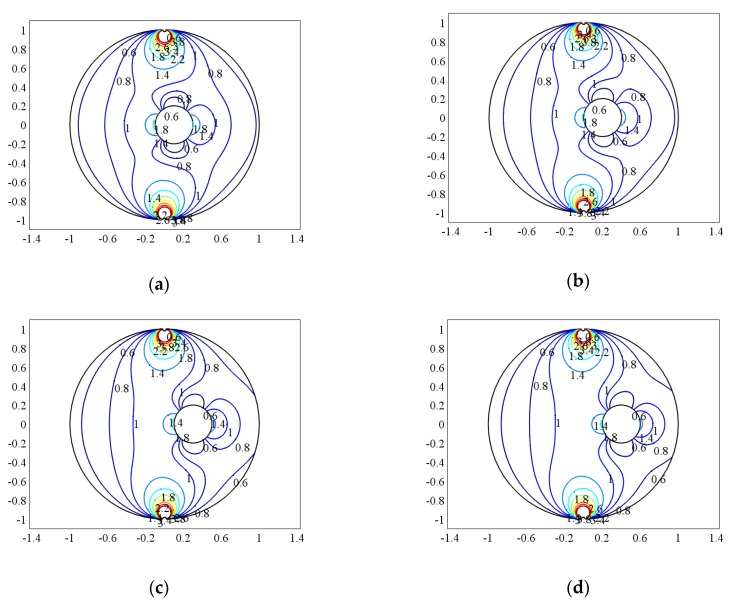
The weight function distribution at a = 0. (**a**) e = 0.1, (**b**) e = 0.2, (**c**) e = 0.3, (**d**) e = 0.4.

**Figure 7 sensors-20-01431-f007:**
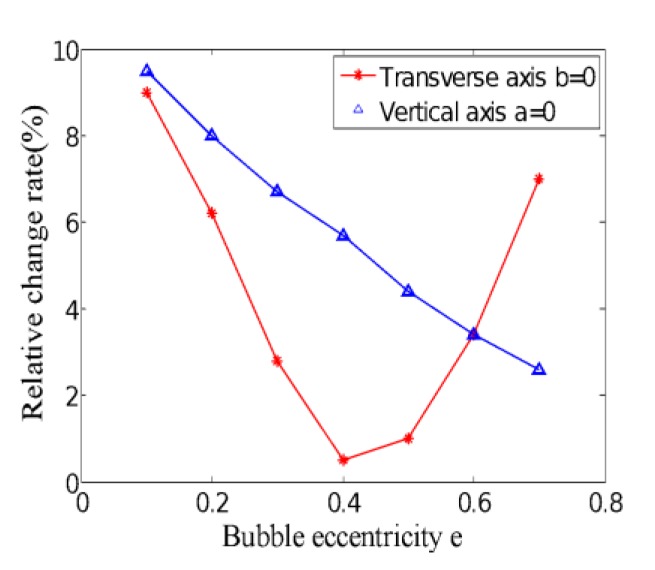
Relative change rate of weight function uniformity at different eccentricity.

**Figure 8 sensors-20-01431-f008:**
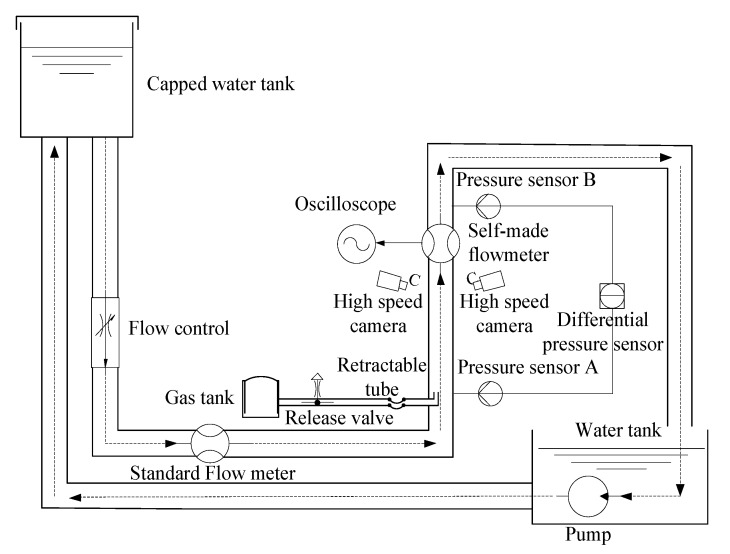
The experimental setup diagram.

**Figure 9 sensors-20-01431-f009:**
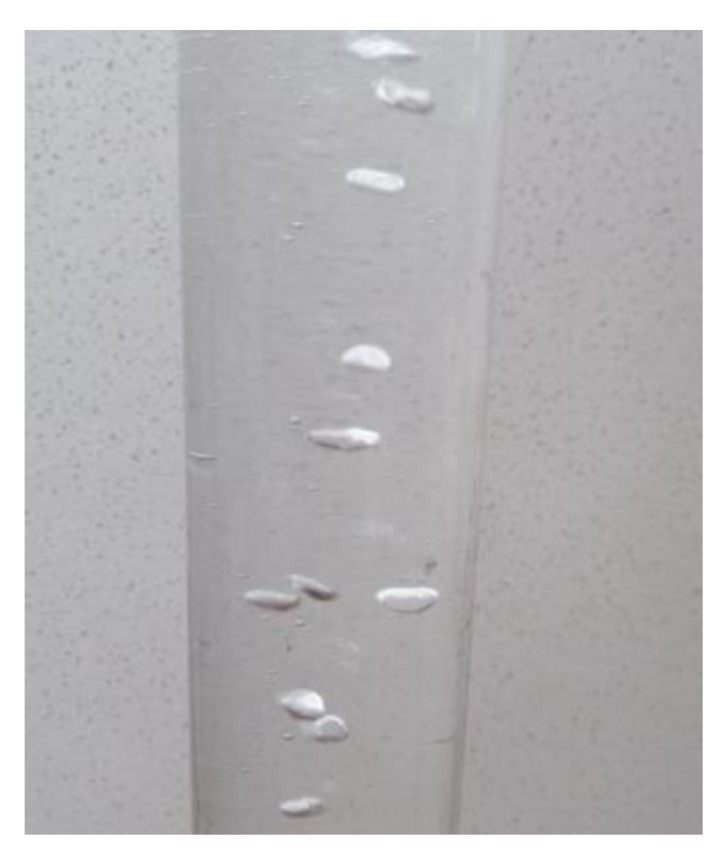
The instantaneous position of the bubbles.

**Figure 10 sensors-20-01431-f010:**
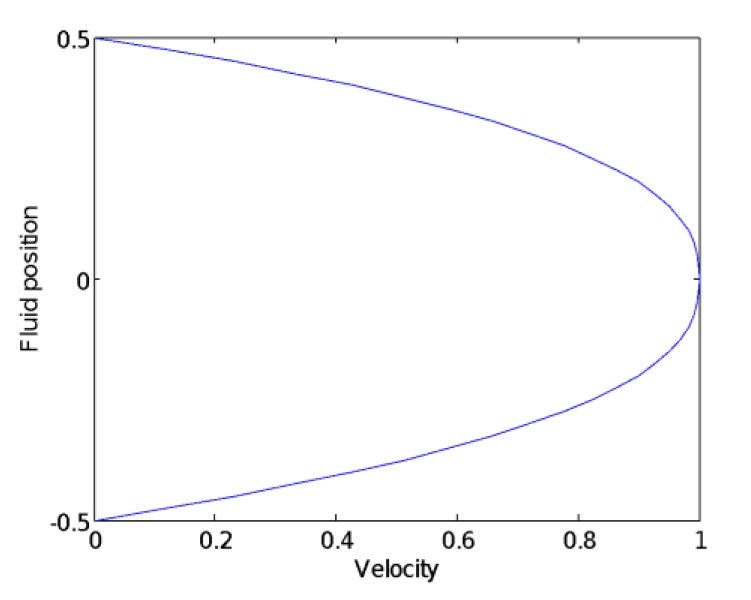
A diagram of the one-dimensional flow velocity distribution.

**Figure 11 sensors-20-01431-f011:**
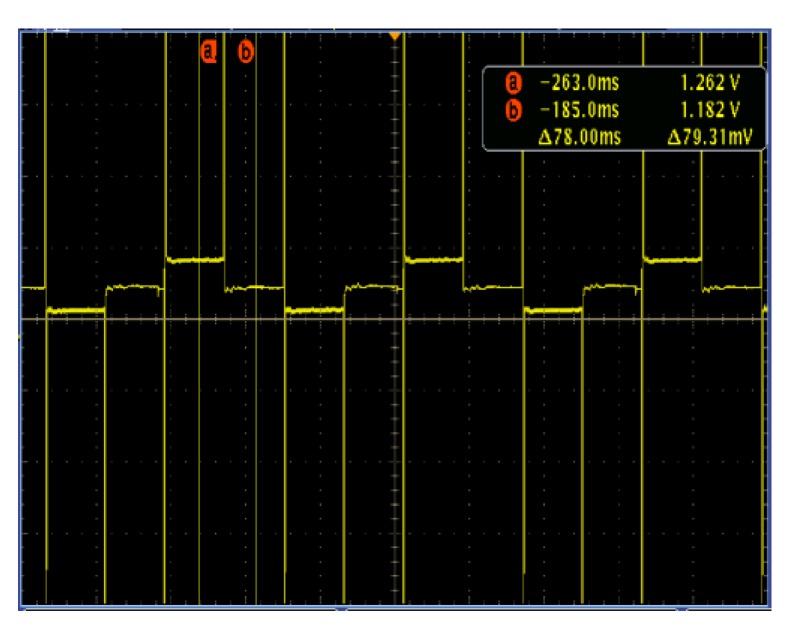
The potential difference between two sensors.

**Table 1 sensors-20-01431-t001:** The uniformity of weight function at concentric.

Bubble Size	q = 0	q = 0.05	q = 0.1	q = 0.15	q = 0.2
Uniformity	0.388	0.386	0.374	0.361	0.345
Relative change rate %	-	0.5	3.6	7.0	11.1

**Table 2 sensors-20-01431-t002:** The potential difference when the bubble is in the center of the pipe.

Bubble Size	q = 0	q = 0.06	q = 0.14	q = 0.23
Bubble position(x,y)(mm)	0	(0.33,0.59)	(0.33,0.59)	(0.33,0.59)
Potential difference(mv)	79.31	79.01	74.47	67.82

**Table 3 sensors-20-01431-t003:** The induced potential difference when the bubble is on the y-axis.

Bubble Position(x,y)(mm)	(0.33,0.59)	(0.16,2.89)	(−0.49,7.12)	(−0.21,9.67)
Potential difference(mv)	67.82	70.27	74.74	76.43

**Table 4 sensors-20-01431-t004:** Induced potential difference when the bubble is on the x-axis.

Bubble Position(x,y)(mm)	(0.33,0.59)	(3.25,0.24)	(6.83,0.08)	(10.13,−0.46)
Potential difference(mv)	67.82	68.57	70.93	65.03

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
