# Peer review of "Study on Weight Function Distribution of Hybrid Gas-Liquid Two-Phase Flow Electromagnetic Flowmeter"

_sensors, 2020, doi:10.3390/s20051431_

Round 1

Reviewer 1 Report

The author has presented a study for calculating the weight function distribution of EMF for investigating the gas-liquid two phase flows. The research is significant in that it employs a traditional single phase flow meter - EMF - in measuring the flow rate of a two-phase flow. The paper can be further improved in various aspects, prior to it being accepted for publication. 

Major comments:

The use of a single phase flow meter (EMF) to measure the two-phase flow is very attractive. As the author pointed out, the weight function distribution of EMF depends on the distribution of gas bubbles, which in turn affects the measurement accuracy. The question is how to obtain the information on the gas holdup/distribution. 

Minor comments:

Figure 1 is confusing and must be improved.  Some notations should be clearly defined, e.g. in Eqs.(7, 9, 28, ...). The definition of source and field should be further explained.

Reviewer 2 Report

The presented research is quite interesting, since it develops the measurement of fluid flow rate of gas-liquid two-phase flows by means of electromagnetic devices.

In order to make the paper compliant with the scopes and aims of the jlournal, the authors should also provide a series of experimental tests aiming at assessing the theoretical considerations made.

Reviewer 3 Report

This is an interesting article on two-phase flow measurements using electromagnetic flowmeters. The authors are recommended to show the range of flow regimes in the two-phase flow map where this method of flow measurement is valid. Clarification is needed on the accuracy of this method and how the accuracy is impacted by a flow with multiple bubbles with a variety of sizes?

Round 2

Reviewer 2 Report

The authors have fulfilled all the suggestions made in the 1st round of reviews. Therefore, the paper can be accepted as is.